# A critical region of A20 unveiled by missense *TNFAIP3* variations that lead to autoinflammation

**Elma El Khouri[1], Farah Diab[1], Camille Louvrier[1,2], Eman Assrawi[1], Aphrodite Daskalopoulou[1], Alexandre Nguyen[3], William Piterboth[2], Samuel Deshayes[3], Alexandra Desdoits[4], Bruno Copin[2], Florence Dastot Le Moal[2], Sonia Athina Karabina[1], Serge Amselem[1,2], Achille Aouba[3†], Irina Giurgea[1,2*†]**

[1]Sorbonne Université, Institut National de la Santé et de la Recherche Médicale (INSERM), "Maladies génétiques d'expression pédiatrique", Paris, France; [2]Département de Génétique Médicale, Hôpital Armand Trousseau, Assistance Publique-Hôpitaux de Paris, Paris, France; [3]Département de Médecine Interne et Immunologie Clinique, Normandie Univ, UNICAEN, UR4650 PSIR, CHU de Caen Normandie, Caen, France; [4]Service de chirurgie pédiatrique, CHU de Caen Normandie, Caen, France

**\*For correspondence:**
irina.giurgea@inserm.fr

†These authors contributed equally to this work

**Competing interest:** The authors declare that no competing interests exist.

**Abstract** A20 haploinsufficiency (HA20) is an autoinflammatory disease caused by heterozygous loss-of-function variations in *TNFAIP3*, the gene encoding the A20 protein. Diagnosis of HA20 is challenging due to its heterogeneous clinical presentation and the lack of pathognomonic symptoms. While the pathogenic effect of *TNFAIP3* truncating variations is clearly established, that of missense variations is difficult to determine. Herein, we identified a novel *TNFAIP3* variation, p.(Leu236Pro), located in the A20 ovarian tumor (OTU) domain and demonstrated its pathogenicity. In the patients' primary cells, we observed reduced A20 levels. Protein destabilization was predicted in silico for A20_Leu236Pro and enhanced proteasomal degradation was confirmed in vitro through a flow cytometry-based functional assay. By applying this approach to the study of another missense variant, A20_Leu275Pro, for which no functional characterization has been performed to date, we showed that this variant also undergoes enhanced proteasomal degradation. Moreover, we showed a disrupted ability of A20_Leu236Pro to inhibit the NF-κB pathway and to deubiquitinate its substrate TRAF6. Structural modeling revealed that two residues involved in OTU pathogenic missense variations (i.e. Glu192Lys and Cys243Tyr) establish common interactions with Leu236. Interpretation of newly identified missense variations is challenging, requiring, as illustrated here, functional demonstration of their pathogenicity. Together with functional studies, in silico structure analysis is a valuable approach that allowed us (i) to provide a mechanistic explanation for the haploinsufficiency resulting from missense variations and (ii) to unveil a region within the OTU domain critical for A20 function.

## Editor's evaluation

The study is valuable to clinical diagnostic laboratories and clinicians as it investigates the functional significance of missense variants in the TNFAIP3 gene encoding A20. Haploinsufficiency of A20 (HA20) is a dominantly inherited disease resulting from monoallelic loss-of-function mutations, while the contribution of novel/rare missense variations is unclear, and they are classified as variants of unknown significance (VUS). The authors developed a new flow cytometry functional assay that other laboratories can adapt to evaluate the pathogenicity of A20 VUSs.

## Introduction

The zinc finger (ZnF) protein A20 is a ubiquitin-editing enzyme encoded by the *TNFAIP3* (*tumor necrosis factor a-induced protein 3*) gene that contains an amino-terminal ovarian tumor (OTU) domain followed by seven ZnF domains. Due to its E3 ligase activity (ZnF4), its deubiquitinase function (OTU) and its ubiquitin-binding domain (ZnF7), A20 regulates the fate of several substrates such as RIPK1 (receptor-interacting serine/threonine-protein kinase 1), NEMO (NF-kappa-B essential modulator), and TRAF6 (TNF receptor-associated factor 6), all converging to the canonical NF-κB pathway (*Boone et al., 2004*; *De et al., 2014*; *Tokunaga et al., 2012*; *Wertz et al., 2004*). Owing to these various functions, A20 plays a pivotal role in the regulation of inflammation and immunity by inhibiting the canonical NF-κB signaling pathway and by preventing cell death (*Lork et al., 2017*; *Martens and van Loo, 2020*). Abnormal expression or function of A20 therefore contributes to the onset of several autoimmune or inflammatory disorders such as Crohn's disease, systemic lupus erythematosus, rheumatoid arthritis, type 1 diabetes mellitus, psoriasis, and atherosclerosis as shown by genome-wide association studies (reviewed in *Malynn and Ma, 2019*). Furthermore, germline variations in *TNFAIP3* lead to an autoinflammatory disease of autosomal dominant inheritance referred to as A20 haploinsufficiency (HA20; MIM#616744), where insufficient suppression of NF-κB activity is observed (*Zhou et al., 2016*). HA20 is a Behçet-like autoinflammatory syndrome of early onset associated with variable clinical signs. The hallmark features of HA20 are recurrent fever, painful oral, genital and/or gastrointestinal ulcers with diarrhea, arthralgia, or polyarthritis (*Aeschlimann and Laxer, 2019*). Since the first description of HA20 (*Zhou et al., 2016*), more than 25 *TNFAIP3* truncating variations have been reported (reviewed in *Yu et al., 2020*) whereas the pathogenic significance has been confirmed for only a part of the 12

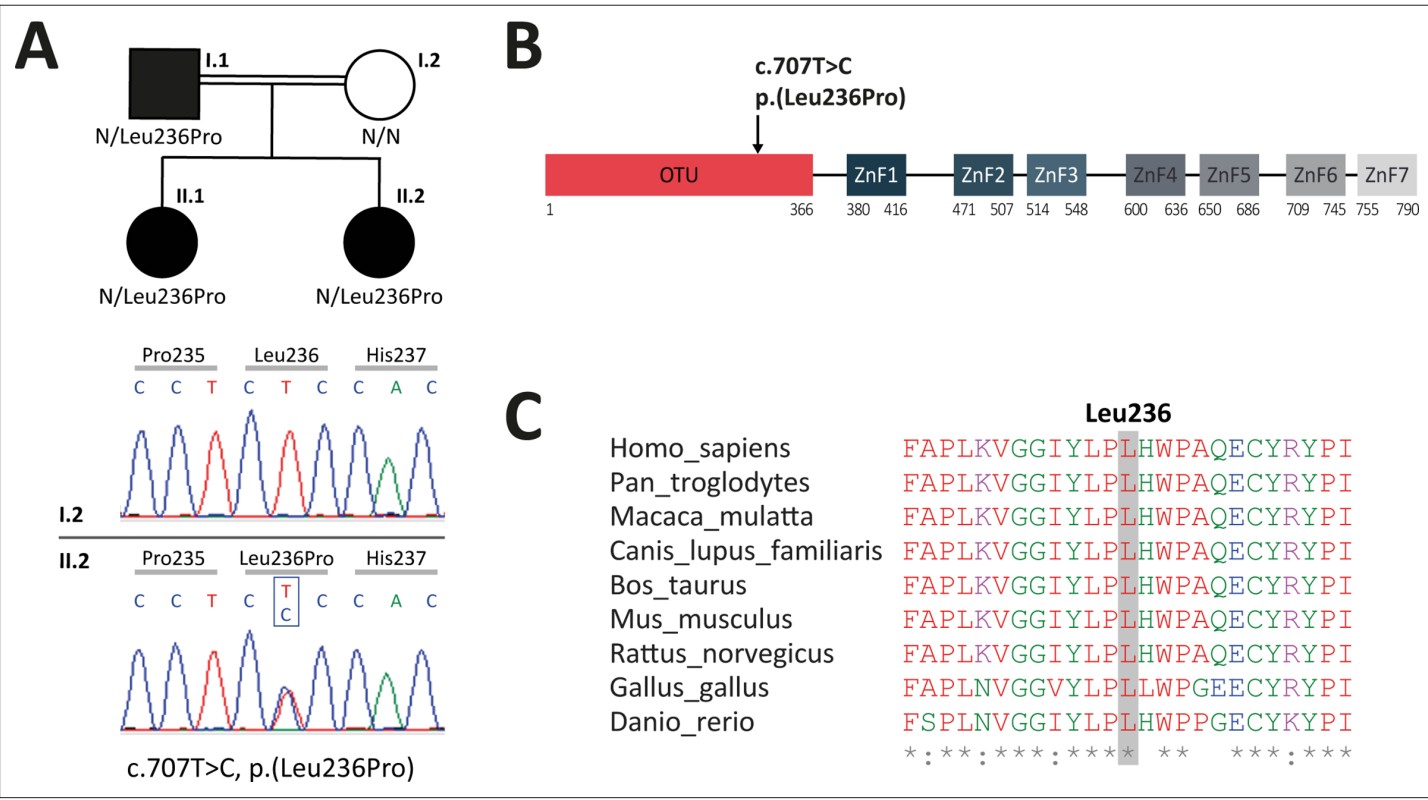

**Figure 1.** Analysis of familial segregation of the c.707T>C, p.(Leu236Pro) variation and its localization on A20 protein. (**A**) Upper panel – genealogical tree of the patients with A20 haploinsufficiency (HA20) (individuals I.1, II.1, and II.2). Lower panel – electropherograms from Sanger sequencing showing the heterozygous c.707T>C, p.(Leu236Pro) variation in individual II.2 but not in her healthy mother (individual I.2). (**B**) Schematic representation of A20 with domain organization showing the ovarian tumor (OTU) domain harboring the p.(Leu236Pro) variation as well as the zinc finger (ZnF) domains. (**C**) Evolutionary conservation of the A20 Leu236 residue across species.

The online version of this article includes the following figure supplement(s) for figure 1:

**Figure supplement 1.** Genealogical tree and electropherograms from Sanger sequencing of the patients with A20 haploinsufficiency (HA20) (individuals I.1, II.1, and II.2) and the healthy individual I.2.

reported missense variations (*Aslani et al., 2022*; *Chen et al., 2020a*; *Dong et al., 2019*; *Jiang et al., 2022*; *Kadowaki et al., 2021*; *Mulhern et al., 2019*; *Niwano et al., 2022*; *Shigemura et al., 2016*; *Tian et al., 2022*). Indeed, a body of genetic and experimental evidence is necessary to determine non-ambiguously the deleterious character of missense variations.

In this study, we report a non-previously described *TNFAIP3* missense variation, c.707T>C, p.(Leu236Pro), identified in three patients from the same family. We provide a phenotypic description of all affected patients, as well as cytokine and A20 level quantification in patient-derived samples. Furthermore, we assessed the pathogenicity of this variant, as well as that of previously reported *TNFAIP3* missense variations, through in vitro molecular and cellular assays. Notably, by studying the location and the intra-molecular amino acid interactions of Leu236 with the other amino acids involved in pathogenic *TNFAIP3* missense variations, we identified a critical region within the OTU domain of A20 where the pathogenic *TNFAIP3* missense variations are in close proximity and where the affected residues share common amino acid interactions on the 3D protein structure.

## Results
### Clinical features of HA20 patients
Proband II.2, a 13-year-old female patient born to a consanguineous union (*Figure 1A*), first experienced autoinflammatory symptoms when she was 15 months of age. She presented with recurrent episodes of fever as well as oral ulcers occurring once every 2 weeks. She suffered from abdominal pain and diarrhea during childhood. She presented with asymmetric arthralgia of large joints and persistent folliculitis. Genital ulcers were very rare. Her sister (individual II.1) was 15 years of age. Her first symptoms, which appeared at the age of 5, included recurrent episodes of fever, persistent oral ulcers, and rare genital ulcers. She also suffered from abdominal pain and diarrhea during childhood. Her skin lesions were acne and folliculitis. Their father (individual I.1) was 38 years of age. His first symptoms, which appeared at the age of 15, included episodes of fever, oral, and genital ulcers occurring once every 2 weeks. He suffered from abdominal pain with diarrhea twice per year and colonic ulcerations were detected at colonoscopy. He also presented with myalgia and arthralgia of the knees or ankles. His skin lesions included folliculitis. He suffered from episcleritis and uveitis at two

**Table 1.** Phenotypic features of individuals I.1, II.1, and II.2 carrying the heterozygous *TNFAIP3* variation c.707T>C (p.Leu236Pro).

| Individual | I.1 | II.1 | II.2 |
|---|---|---|---|
| Gender | M | F | F |
| Current age | 38 | 15 | 13 |
| Age at onset | 15 years | 5 years | 15 months |
| Recurrent fever | Yes | Yes | Yes |
| CRP (mg/L) | 15 | 7 | 7 |
| Ulcers | Oral, genital | Oral, genital (rare) | Oral, genital (rare) |
| Gastrointestinal | Abdominal pain, diarrhea, colonic ulcerations | Abdominal pain, diarrhea | Abdominal pain, diarrhea |
| Musculoskeletal | Myalgia – oligoarthralgia (knee or ankle) | No | Asymmetric oligoarthralgia (large joints) |
| Skin | Folliculitis | Acne, folliculitis | Folliculitis |
| Ocular | Episcleritis – anterior uveitis | No | No |
| Cardiovascular | No | No | No |
| Hepatic | No | No | No |
| Treatment | Colchicine – corticosteroids – azathioprine | Colchicine | Colchicine |

CRP: C-reactive protein

occasions. Individuals II.1 and II.2 show significant improvement of their symptoms upon colchicine treatment. Individual I.1 received colchicine treatment and then tapered courses of oral corticosteroids and azathioprine, leading to a satisfactory long-term control of the disease. The clinical features of the three patients are summarized in *Table 1*.

## Identification of a novel missense variation in *TNFAIP3*

To identify the molecular basis of the patients' disease, targeted next-generation sequencing (NGS) for autoinflammatory diseases was conducted on a DNA sample from proband II.2. It revealed a non-previously reported heterozygous variation c.707T>C, p.(Leu236Pro) in the *TNFAIP3* gene. Noteworthy, in this patient born to a consanguineous union, no mutation was found in the following genes known to be involved in autosomal recessive autoinflammatory diseases: *ADA2, IL10, IL10RA, IL10RB, IL1RN, IL36RN, IL6ST, LACC1, LPIN2, MEFV, MVK, NLRP1, OTULIN,* and *TRNT1*. Sanger sequencing confirmed the *TNFAIP3* c.707T>C, p.(Leu236Pro) variation in the proband and identified this variation in the related affected individuals (I.1 and II.1) (*Figure 1A*, *Figure 1—figure supplement 1*). According to the American College of Medical Genetics and Genomics (ACMG) guidelines to assess the pathogenicity of sequencing variations, the c.707T>C, p.(Leu236Pro) variation is of uncertain significance. However, the affected leucine residue at position 236 (Leu236) is located in the OTU domain of A20 (*Figure 1B*) and is invariant across species (*Figure 1C*), thereby supporting a deleterious effect of this amino acid substitution. The c.707T>C, p.(Leu236Pro) variant has not been described in sequence variant databases, such as the genome aggregation database (gnomAD, https://gnomad.broadinstitute.org). According to the Combined Annotation-Dependent Depletion (CADD, https://cadd.gs.washington.edu/snv) tool, the c.707T>C, p.(Leu236Pro) variation displays a score of 29.5, predictive of a pathogenic variation.

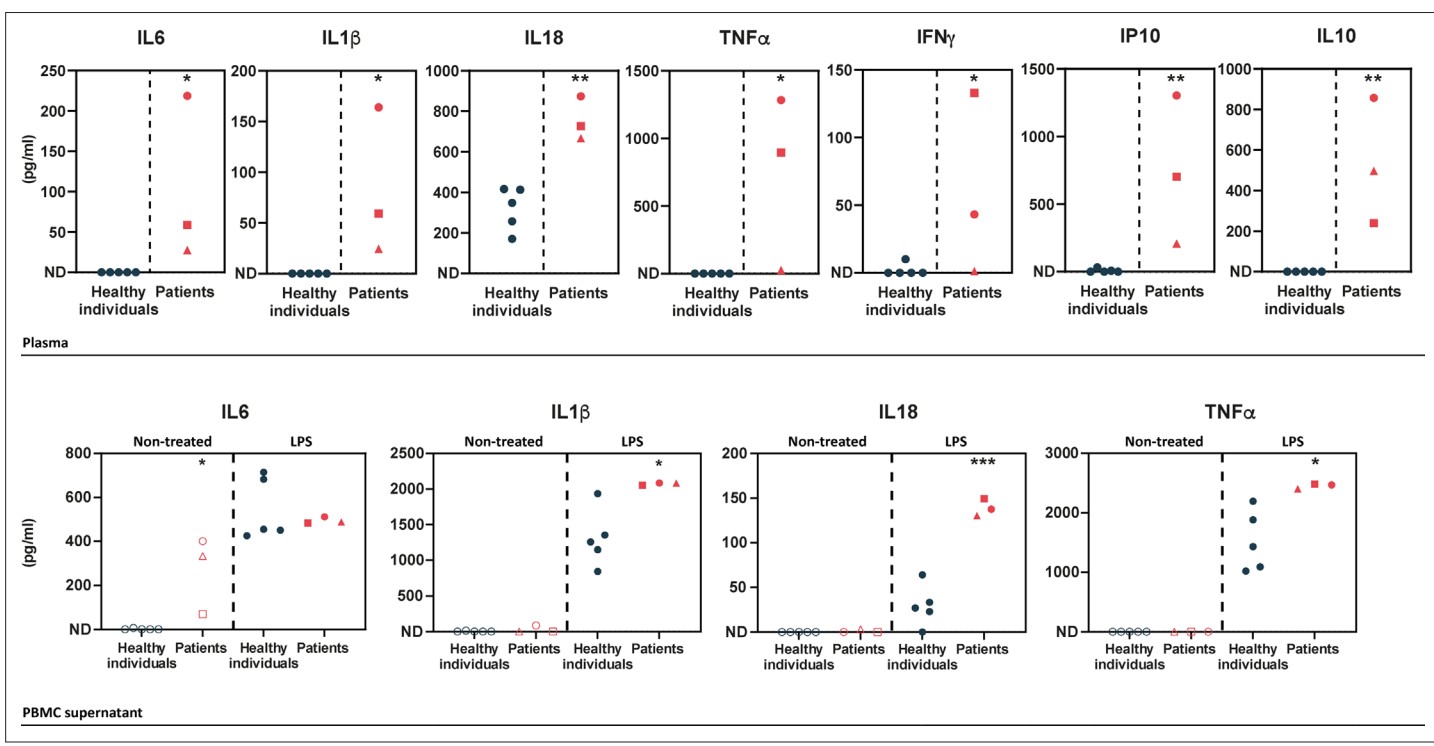

**Figure 2.** Upper panel – ELISA-assessed plasma cytokine levels in the patients (individual I.1: red square, individual II.1: red triangle, and individual II.2: red circle) and five healthy donors (blue circles). Lower panel – ELISA-assessed cytokine levels in peripheral blood mononuclear cell (PBMC) supernatants prior to (empty symbols) and upon (filled symbols) LPS treatment (100 ng/mL for 16 hr) (individual I.1: red square, individual II.1: red triangle, and individual II.2 red circle, and five healthy donors: blue circles). ND: not detected. Unpaired two-tailed Student's t-test was used and asterisks indicate that the mean of the cytokine levels quantified in samples from five healthy individuals is significantly different from the mean of the cytokine levels quantified in samples from the three patients. p-Values <0.05 were considered statistically significant. p-Values <0.05, <0.01, and <0.001 are indicated with *, **, and ***, respectively.

## Cytokine profile in patients-derived PBMCs

In order to determine the cytokine profile of the patients identified in this study, we quantified inflammatory cytokines in plasma samples and in peripheral blood mononuclear cell (PBMC) supernatants upon LPS cell treatment. Plasma levels of pro-inflammatory cytokines (IL1β, IL18, IL6, TNFα, IFNγ, and IP10) were substantially elevated in the patients (*Figure 2* – upper panel), as compared to healthy individuals. The patients also exhibited high levels of the anti-inflammatory cytokine IL10. In LPS-stimulated PBMCs, pro-inflammatory cytokine levels were high in the patients, as compared to healthy individuals (*Figure 2* – lower panel) except for IL6 which exhibited high levels in the patients' samples prior to stimulation. IFNγ, IP10, and IL10 cytokines were not detected in the PBMC supernatants of healthy individuals or HA20 patients.

## Evidence for HA20 from patients-derived samples

To assess the pathogenicity of the p.(Leu236Pro) variation, we first sought to analyze A20 protein expression in patients' PBMCs. In contrast to two healthy individuals, all three patients (individuals I.1, II.1, and II.2) exhibited a significant reduction of A20 expression levels (*Figure 3A*).

## Decreased stability of the A20_Leu236Pro protein

With the aim of determining the molecular basis of reduced A20 protein expression in the patients and further assessing the impact of the p.(Leu236Pro) missense variation on the protein, we performed in silico stability analyses of the available human crystal structure of the OTU domain of A20 in its active form (Protein Data Bank: 3ZJD) (*Kulathu et al., 2013*) using the bioinformatics tool PremPS (*Chen et al., 2020b*). The single variation Leu236Pro of A20 is expected to be destabilizing, as indicated by the positive value of 1.820 kcal/mol for the change of unfolding Gibbs free energy ΔΔG (*Table 2*). Similar results were obtained using two other prediction tools: Site Directed Mutator (ΔΔG=1.860 kcal/mol) and MAESTRO (Multi AgEnt STability pRedictiOn) (ΔΔG=1.752 kcal/mol).

To confirm these in silico results, we assessed the expression and stability of the A20_Leu236Pro protein in HEK293T cells transiently expressing A20_WT or A20_Leu236Pro. As shown in *Figure 3B*, the steady-state protein amounts of A20_Leu236Pro were only 47.6% of that of the WT protein (0.54 for A20_WT vs 0.26 for A20_Leu236Pro). To determine if this difference resulted from protein instability, we compared the half-life of each protein upon inhibition of protein synthesis with cycloheximide in HEK293T cells. Quantification of protein amounts at different time points showed that the half-life of A20_Leu236Pro was 2.8 hr, whereas the A20_WT protein was still highly expressed even after 24 hr of treatment (*Figure 3C*). This substantial reduction of the protein half-life shows that A20_Leu236Pro is less stable than its WT counterpart. To test whether this reduced stability was associated with abnormal protein degradation by the ubiquitin-proteasome system (UPS), we inhibited the proteasome-dependent degradation with the peptide-aldehyde inhibitor MG132 in HEK293T cells. As shown in *Figure 3D*, substantial accumulation of A20_Leu236Pro was observed upon inhibition of the proteasome up to 91.3% of the WT protein levels (2.52 for A20_WT vs 2.30 for A20_Leu236Pro). This strongly suggests that the difference of the amounts of WT and Leu236Pro A20 proteins in non-treated cells results from enhanced UPS-dependent degradation.

## Decreased inhibition of NF-κB activity by the A20_Leu236Pro protein

Given the role of A20 in the regulation of the NF-κB activity, we assessed the impact of the p.(Leu236Pro) variation on the ability of A20 to inhibit the canonical NF-κB pathway. To this end, we transiently transfected HEK293T cells with an empty vector, A20_WT or A20_Leu236Pro encoding plasmids along with NF-κB reporter vector and Renilla vector for normalization, and then quantified the luminescence signal upon TNFα treatment (10 ng/mL, 5 hr). Compared to an empty vector, A20_WT exhibited up to eightfold decrease of the NF-κB activity in a dose-dependent manner (*Figure 3E*). A20_Leu236Pro failed to suppress TNFα-induced NF-κB activity up to the levels of the WT counterpart: at 25 ng, A20_Leu236Pro decreased the NF-κB activity only by two folds compared to the empty vector. Moreover, despite doubling the doses of A20_Leu236Pro (50 ng), the level of NF-κB inhibition observed upon A20_WT expression was not reached (*Figure 3E*).

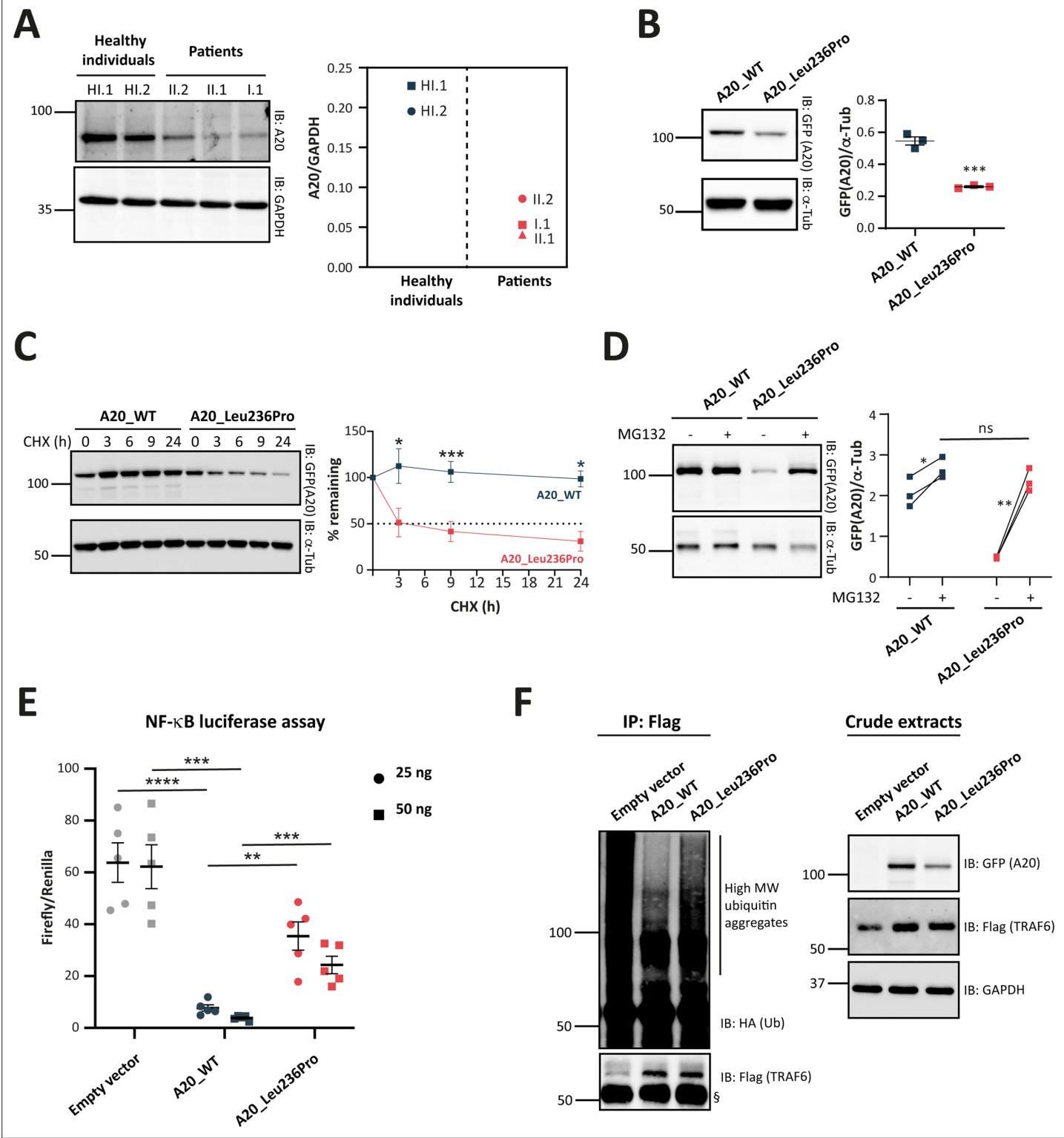

**Figure 3.** Evaluation of the pathogenicity of the A20_Leu236Pro variation through cell-based assays. (**A**) A20 protein expression in peripheral blood mononuclear cells (PBMCs) of healthy individuals and individuals I.1, II.1, and II.2. (**B**) Steady-state protein levels of A20_WT and A20_Leu236Pro upon transient expression in HEK293T (n=3, p-value = 0.0004). (**C**) Protein levels of A20_WT and A20_Leu236Pro upon transient expression in HEK293T and treatment with cycloheximide (100 µg/mL) at the indicated time points. (n=3, p-value = 0.0158, 0.0008, and 0.026 for time points 3, 9, and 24 hr, respectively). (**D**) Protein levels of A20_WT and A20_Leu236Pro upon transient expression in HEK293T and treatment with MG132 (20 µM – 5 hours) (+) or without MG132 (-) and with DMSO (n=3, p-values for (A20_WT-DMSO vs -MG132), (A20_Leu236Pro-DMSO vs -MG132), and (A20_WT-MG132 vs

*Figure 3 continued on next page*

*Figure 3 continued*

A20_Leu236Pro-MG132)=0.037, 0.006, and 0.125, respectively. (**E**) Quantification of the NF-$\kappa$B signaling in HEK293T cells transiently expressing 25 or 50 ng of empty vector, A20_WT or A20_Leu236Pro. Cells were treated with 10 ng/mL TNF$\alpha$ for 5 hr. Firefly luciferase activity was normalized to the Renilla signal (n=5), p-values for (A20_WT vs Empty vector – 25 ng), (A20_WT vs A20_Leu236Pro – 25 ng), (A20_WT vs Empty vector – 50 ng), (A20_WT vs A20_Leu236Pro – 50 ng) are <0.0001, = 0.0011, = 0.0001, and = 0.0003, respectively. (**F**) Ubiquitination profile of A20 target TRAF6 (flag-tagged) upon co-expression with GFP-vector, GFP-tagged A20_WT, or A20_Leu236Pro together with HA-K63Ub. Flag-TRAF6 was immunoprecipitated from protein lysates using the ANTI-FLAG M2 Magnetic beads and high molecular weight ubiquitin aggregates were revealed by immunoblotting of precipitates with HA-specific antibody. The § sign indicates the antibody heavy chain. The expression of the proteins in the crude extracts is shown to the right. This experiment is representative of a set of three experiments performed independently. For all western blot experiments, equal amounts of protein extracts were subjected to SDS-PAGE and immunoblotted with the indicated antibodies. The A20 or GFP (the TNFAIP3-expressing plasmid used is pEGFP-C1-A20) signal was quantified with ImageJ software and normalized to the amount of GAPDH or $\alpha$-tubulin used as a loading control. For (**A, B, C, D, and E**), unpaired two-tailed Student's t-test was used and asterisks indicate that the mean is significantly different; p-values <0.05 were considered statistically significant. p-Values <0.05, <0.01, <0.001, and <0.0001 are indicated with *, **, ***, and ****, respectively. Data are plotted with SEM error bars.

The online version of this article includes the following source data for figure 3:

**Source data 1.** A20 protein expression in peripheral blood mononuclear cells (PBMCs) of healthy individuals and individuals I.1, II.1, and II.2.

**Source data 2.** Protein levels of A20_WT and A20_Leu236Pro upon transient expression in HEK293T.

**Source data 3.** Protein levels of A20_WT and A20_Leu236Pro upon transient expression in HEK293T and treatment with cycloheximide.

**Source data 4.** Protein levels of A20_WT and A20_Leu236Pro upon transient expression in HEK293T and treatment with MG132.

**Source data 5.** Ubiquitination profile of Flag-TRAF6 upon expression of A20_WT or A20_Leu236Pro.

## Decreased deubiquitination activity of the A20_Leu236Pro protein

To investigate whether the A20_Leu236Pro still retained its deubiquitinase (DUB) function, we co-expressed A20_WT or A20_Leu236Pro with K63-linked ubiquitin and the A20 target TRAF6, then evaluated the ubiquitination state of TRAF6. This co-immunoprecipitation experiment revealed an increased intensity of the high molecular weight ubiquitin aggregate smear of TRAF6 in the presence of A20_Leu236Pro compared to the WT condition, thus indicating an altered DUB activity of A20_Leu236Pro (*Figure 3F*).

## Review of all previously reported *TNFAIP3* missense variations

To better assess the impact of the localization of *TNFAIP3* missense variations on A20 protein function, we reviewed all missense variations reported to date that are shared with the scientific community as original articles or conference abstracts, including the variation herein identified (*Table 2*). The majority (7/13) are located in the OTU domain: c.707T>C, p.(Leu236Pro) (this study), c.305A>G, p.(Asn102Ser) (*Chen et al., 2020a*), c.386C>T, p.(Thr129Met) (*Aslani et al., 2022*), c.574G>A, p.(Glu192Lys) (*Kadowaki et al., 2021*), c.728G>A, p.(Cys243Tyr) (*Shigemura et al., 2016*), c.824T>C, p.(Leu275Pro) (*Aslani et al., 2022*), and c.929T>C, p.(Ile310Thr) (*Kadowaki et al., 2021*). Other missense variations have been reported outside the OTU domain: c.1129G>A, p.(Val377Met) (*Niwano et al., 2022*), c.1428G>A, p.(Met476Ile) (*Dong et al., 2019*), c.1639G>A, p.(Ala547Thr) (*Tian et al., 2022*), c.1804A>T, p.(Thr602Ser) (*Jiang et al., 2022*), c.1939A>C, p.(Thr647Pro) (*Mulhern et al., 2019*), and c.2126A>G, p.(Gln709Arg) (*Kadowaki et al., 2021*), which are located between OTU and ZnF1, in ZnF2, in ZnF3, in ZnF4, between ZnF4 and ZnF5, and in ZnF6, respectively (*Table 2*).

As for the missense variations located in the OTU domain, the pathogenic significance is demonstrated for four of them: p.(Glu192Lys) (*Kadowaki et al., 2021*), p.(Leu236Pro) (reported in this study), p.(Cys243Tyr) (*Shigemura et al., 2016*), and p.(Leu275Pro) (*Aslani et al., 2022*) and, which all fulfill the ACMG criteria for pathogenicity (*Richards et al., 2015*). To date, no functional study has been carried out for the A20_Leu275Pro variant, but since it is a de novo variation, absent from gnomAD, with a high CADD score (31) and involving a highly conserved residue across species, it meets ACMG criteria for pathogenicity. The remaining missense variations located in the OTU domain, p.(Asn102Ser), p.(Thr129Met), and p.(Ile310Thr), cannot be classified as pathogenic according to ACMG criteria. In fact, the p.(Asn102Ser) variation has an allele frequency in the general population gnomAD database of 3482/281824 (0.01236), which is incompatible with the rare occurrence of HA20; we therefore classified this variation as non-pathogenic despite the potential localization of Asn102 residue within a pocket in the active site of A20 that is important for its DUB activity (*Komander and Barford, 2008*;

**Table 2.** Patients' phenotype and molecular features associated with the c.707T>C, p.(Leu236Pro) and all the *TNFAIP3* missense variations reported in the literature.

| Molecular TNFAIP3 (NM_006290) anomaly | c.707T>C, p.(Leu236Pro) | c.305A>G, p.(Asn102Ser) | c.386C>T, p.(Thr129Met) | c.574G>A, p.(Glu192Lys) | c.728G>A, p.(Cys243Tyr) | c.824T>C, p.(Leu275Pro) | c.929T>C, p.(Ile310Thr) | c.1129G>A, p.(Val377Met) | c.1428G>A, p.(Met476Ile) | c.1639G>A, p.(Ala547Thr) | c.1804A>T, p.(Thr602Ser) | c.1939A>C, p.(Thr647Pro) | c.2126A>G, p.(Gln709Arg) |
|---|---|---|---|---|---|---|---|---|---|---|---|---|---|
| **Reference** | Current study | Chen et al., 2020b | Aslani et al., 2022 | Kadowaki et al., 2021 | Shigemura et al., 2016 | Aslani et al., 2022 | Kadowaki et al., 2021 | Niwano et al., 2022 | Dong et al., 2019 | Tian et al., 2022 | Jiang et al., 2022 | Mulhern et al., 2019 | Kadowaki et al., 2021 |
| **Domain** | OTU | OTU | OTU | OTU | OTU | OTU | OTU | Between OTU and Znf1 | Znf2 | Znf3 | Znf4 | Between Znf4 and Znf5 | Znf6 |
| **Age at onset** | 15 months | 15 years | 4 months | 11 months | 9 years | 3 months | 12 years | 20 years | 13 years | 18 years | Before 32 months | 6 months | 12 years |
| **Recurrent fever** | Yes | No | Yes§ | Yes | Yes | Yes | Yes | Yes | Yes | Yes | Yes | No | No |
| **Ulcers** | Oral and genital (rare) | Oral and gastrointestinal | No | Aphthous stomatitis | Oral, genital and gastrointestinal | No | Gastrointestinal | No | Oral and gastrointestinal | Oral and genital | Oral, genital, and gastrointestinal | Oral (occasional) | No |
| **Family segregation** | Sister and father carrying the variation and presenting HA20 symptoms | Father and brother carry the variation and suffer from recurrent fever and oral ulcers – milder symptoms | No family history | Father carries the variation with recurrent stomatitis, folliculitis, and hemorrhoids | Mother and three other family members carry the variation and suffer from ulcers | No family history – de novo variation | Mother carries the variation, no symptoms | Mother and brother with same symptoms – no genetic analysis | Mother carries the variation and shows milder symptoms | Mother and sisters carry the variation and suffer from isolated oral ulcers | Mother and brother carry the variation and suffer from oral ulcers | Mother and sister carry the variation and are (mildly) symptomatic | Father carries the variation, no symptoms |
| **HA20 hallmark features of the index case** — Treatment | Colchicine (partial response) | Glucocorticoids and thalidomide (less frequent and less severe symptoms) | Etoposide, oral dexamethasone, and cotrimoxazole (deceased from cerebral hemorrhage) | Corticosteroids (variable response) | Colchicine (unsuccessful) Corticosteroids (successful) | Anti-TNFα (successful) | Colchicine (no response), NSAID's and corticosteroids (partial response), infliximab +methotrexate (successful) | Colchicine (no response), corticosteroids +TNFα inhibitors (successful) | Acyclovir and corticosteroids (transient improvement) | Corticosteroids, methotrexate, and thalidomide. Tacrolimus and leflunomide (successful) | Colchicine and corticosteroids | Corticosteroids and immunosuppressors (cyclophosphamide) (unsuccessful) – JakI/2 inhibitors (considerable improvement) | Corticosteroids |
| **GnomAD (v2.1.1)** | Absent | 3482/281824 (6.3% in Latino population) | 4/282896 | Absent | Absent | Absent | 3/282356 | 10/251,490 | Absent | Absent | Absent | 511/282864 | Absent |
| **CADD score (GRCh37-v1.6)** | 29.5 | 25.8 | 25.8 | 32 | 25.9 | 31 | 22.6 | 0.001 | 28 | 8.432 | 2.206 | 14.1 | 14.12 |
| **Conservation** | Highly conserved | Highly conserved | Partially conserved | Highly conserved | Highly conserved | Highly conserved | Partially conserved | Poorly conserved (met in several species) | Highly conserved | Poorly conserved | Partially conserved | Poorly conserved | Partially conserved |
| **PremPS predicted ΔΔG (kcal/mol)** | 1.82 | 0.78 | −0.38 | 0.65 | 0.05 | 1.80 | 1.01 | na | na | na | na | na | na |
| **Functional consequence** | Reduced A20 expression (in vitro and patient's samples) and reduced NF-κB inhibition, decrease of deubiquitinase activity (in vitro)* | | | Reduced NF-κB inhibition (in vitro)* | Reduced A20 expression and reduced NF-κB inhibition (in vitro)* | Reduced A20 expression (in vitro) data from the current study | Reduced A20 expression – Inhibition of NF-κB was comparable to that of the wild-type (in vitro)* | Enhanced phosphorylation of NF-κB p65, reduced A20 expression, and attenuated phosphorylation of A20 in patient's samples* | Decreased A20 expression, NF-κB pathway overactivation and synthesis of TNF-α was upregulated in patient-derived cells* | | Increased IκBα degradation and p65 phosphorylation in patient's samples and in vitro (THP-1 cells)* | Altered NEMO ubiquitination in patients' samples – reduced NF-kB inhibition (in vitro) (Kadowaki et al., 2021)* | Inhibition of NF-κB was comparable to that of the wild-type (in vitro)* |
| **Variation characteristics** — Pathogenic significance§ | Likely pathogenic | Benign | Uncertain significance | Likely pathogenic | Likely pathogenic | Pathogenic | Uncertain significance | Uncertain significance | Uncertain significance | Uncertain significance | Likely pathogenic | Likely benign | Likely benign |

GnomAD: Genome Aggregation Database, CADD: Combined Annotation-Dependent Depletion, na: not applicable.

§: according to the papers reporting the variation: c.305A>G, p.(Asn102Ser) (*Chen et al., 2020a*), c.386C>T, p.(Thr129Met) and c.824T>C, p.(Leu275Pro) (*Aslani et al., 2022*), c.574G>A, p.(Glu192Lys), c.929T>C, p.(Ile310Thr) and c.2126A>G, p.(Gln709Arg) (*Kadowaki et al., 2021*), c.728G>A, p.(Cys243Tyr) (*Shigemura et al., 2016*), c.1428G>A, p.(Met476Ile) (*Dong et al., 2019*), c.1129G>A, p.(Val377Met) (*Niwano et al., 2022*), c.1639G>A, p.(Ala547Thr) (*Tian et al., 2022*), c.1804A>T, p.(Thr602Ser) (*Jiang et al., 2022*) and c.1939A>C, p.(Thr647Pro) (*Mulhern et al., 2019*); §: according to the ACMG criteria for pathogenicity (*Richards et al., 2015*).

GnomAD: Genome Aggregation Database, CADD: Combined Annotation-Dependent Depletion, na: not applicable.

*according to the papers reporting the variation: c.305A>G, p.(Asn102Ser) (*Chen et al., 2020a*), c.386C>T, p.(Thr129Met) and c.824T>C, p.(Leu275Pro) (*Aslani et al., 2022*), c.574G>A, p.(Glu192Lys), c.929T>C, p.(Ile310Thr) and c.2126A>G, p.(Gln709Arg) (*Kadowaki et al., 2021*), c.728G>A, p.(Cys243Tyr) (*Shigemura et al., 2016*), c.1428G>A, p.(Met476Ile) (*Dong et al., 2019*), c.1129G>A, p.(Val377Met) (*Niwano et al., 2022*), c.1639G>A, p.(Ala547Thr) (*Tian et al., 2022*), c.1804A>T, p.(Thr602Ser) (*Jiang et al., 2022*) and c.1939A>C, p.(Thr647Pro) (*Mulhern et al., 2019*).

*Lin et al., 2008*). The p.(Thr129Met) variation can be considered as a variation of uncertain significance as the Thr129 residue is only partially conserved and no functional assays support its pathogenic significance (*Aslani et al., 2022*). As for the p.(Ile310Thr) variation, it was qualified by the authors of uncertain significance due to the lack of evolutionary conservation of the affected amino acid together with the absence of functional alteration of the variant (*Kadowaki et al., 2021*; *Table 2*).

Among the six *TNFAIP3* missense variations located outside the OTU domain, only the p.(Thr602Ser) variation can be considered as pathogenic according to ACMG criteria. The variations p.(Val377Met), p.(Met476Ile), and p.(Ala547Thr) are considered of uncertain significance as no in vitro functional test was performed to assess their pathogenicity. The allele frequency of p.(Thr647Pro) in gnomAD is 511/282864 and the individuals carrying this variation lack HA20 hallmark clinical features (*Mulhern et al., 2019*), we therefore considered it as likely benign (*Table 2*). As for the p.(Gln709Arg) variation, it is considered as likely benign by the authors due to the lack of evolutionary conservation of Gln709 and the absence of protein function alteration (*Kadowaki et al., 2021*; *Table 2*).

## Functional validation of in silico protein destabilization prediction of different A20 missense variants

In order to test whether *TNFAIP3* missense variations other than the p.(Leu236Pro) variant exhibit enhanced proteasomal degradation, we sought to quantify the levels of several recombinant A20 variant proteins, upon MG132-induced proteasome inhibition. For this purpose, we took advantage of the green fluorescent protein (GFP) tag present in the *TNFAIP3* constructs to sort the cells by flow cytometry and quantify the intensity of the GFP signal in cells treated with MG132 or DMSO as negative control. For this analysis, in addition to A20_WT and A20_Leu236Pro, we included the likely pathogenic p.(Leu275Pro) variant, described in Aslani et al. because the in silico PremPS tool predicted a high destabilizing effect of the variation ($\Delta\Delta G=1.80$ kcal/mol – *Table 2*) and Leu275Pro protein expression has not been assessed to date. As positive control, we used the construct coding for the A20_Leu277* truncated protein (*Zhou et al., 2016*) which should be recognized by the protein quality control as a misfolded protein. We also chose to include the p.(Asn102Ser) variant, previously suggested to lead to HA20 (*Chen et al., 2020a*), while keeping in mind that it is frequent in the control populations (allele frequency of 0.01236 if considering all populations and of 0.06389 in the Latino population according with gnomAD) and has a low destabilizing predicted effect ($\Delta\Delta G=0.78$ kcal/mol – *Table 2*). As negative control, we chose the polymorphic variant p.(Phe127Cys) (allele frequency = 0.0617 in gnomAD). Fluorescence intensity increased for A20_Leu236Pro, A20_Leu275Pro and the positive control A20_Leu277*, indicating that like Leu236Pro, the Leu275Pro variant exhibits enhanced proteasomal degradation (curve shifted to the right in *Figure 4* upper panel and increased fluorescence intensity in *Figure 4* lower panel). This observation is specific to pathogenic variants as no fluorescence intensity shift was observed for the non-pathogenic p.(Phe127Cys) variant nor for the p.(Asn102Ser) variant in the presence of MG132. This latter result, which demonstrates the absence of enhanced proteosomal degradation of the A20_Asn102Ser protein, confirms the classification of this variant as non-pathogenic (*Figure 4*).

## Identification of an OTU subdomain containing three *TNFAIP3* pathogenic missense variations

The positioning of residues affected by the pathogenic missense variations on the human crystal of A20 OTU domain revealed that Glu192 and Cys243 and Leu275 are in close vicinity to Leu236, suggesting possible common interactions (*Figure 5A*). To address this hypothesis, we first used the PremPS prediction tool to model the interactions established by each of these residues within the WT protein and to predict the impact of each variation on these interactions (*Figure 5B, C and D*). Additionally, we sought to identify overlapping interaction alterations between these four variations. This analysis revealed common interactions between Leu236, Glu192, and Cys243, but not with Leu275. We therefore focused the amino acid interactions analysis on residues Glu192, Leu236, and Cys243 (*Figure 5B–G*). In A20_WT, Leu236 establishes several hydrogen and/or polar bonds as well as hydrophobic interactions with the surrounding residues; some of these interactions are altered in A20_Leu236Pro, for example, interactions with Phe197, Asn201, and Tyr306 (*Figure 5B*). Modified interactions are also observed in A20_Glu192Lys where Lys192 establishes additional interactions with Leu171 and Met174 compared to Glu192 (*Figure 5C*) and particularly in A20_Cys243Tyr where the

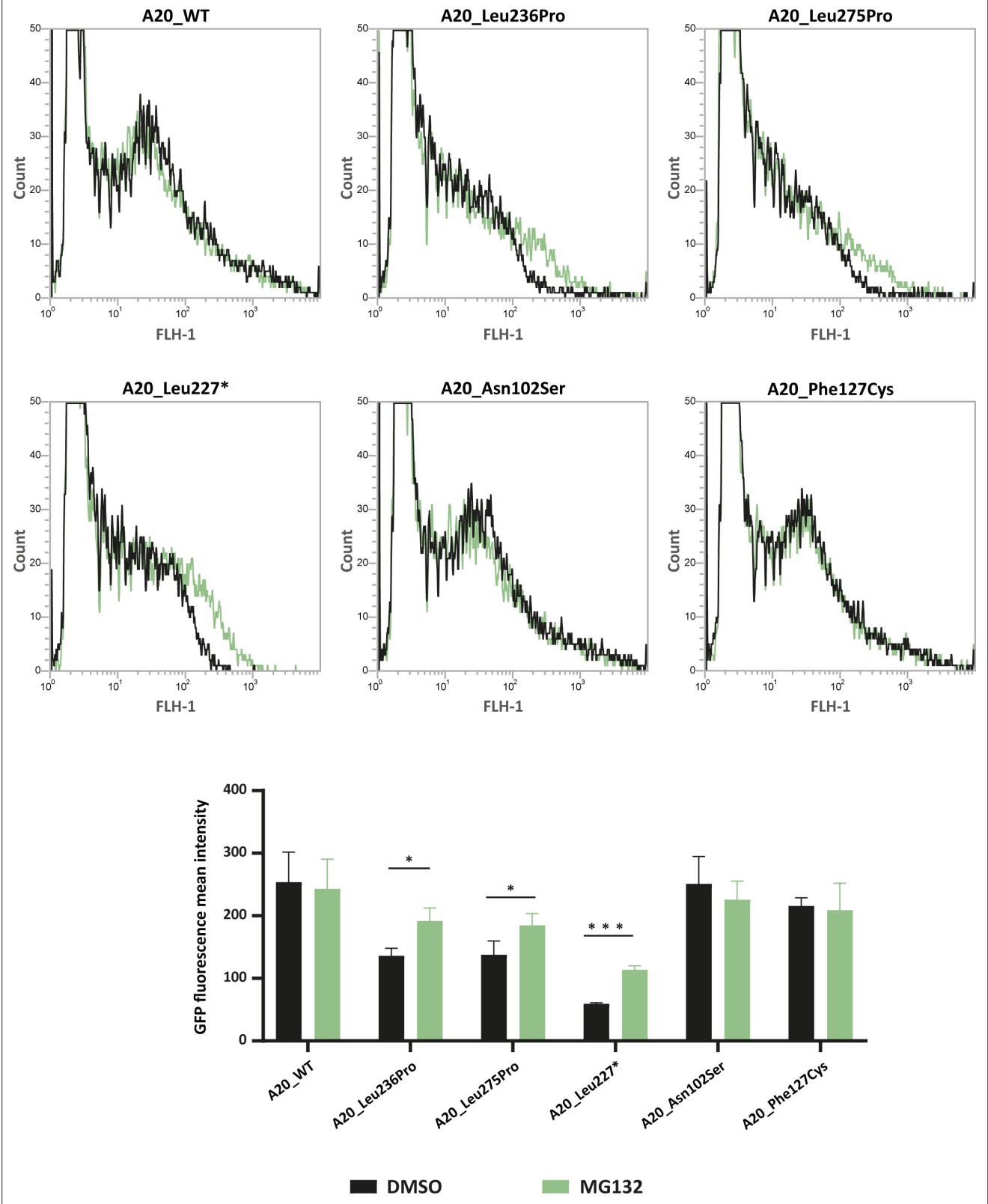

**Figure 4.** Flow cytometry quantification of green fluorescent protein (GFP) intensity in protein extracts from HEK293T cells expressing GFP-tagged A20_WT or A20 variants treated with DMSO or MG132 (20 μM – 5 hr). Representative fluorescence-activated cell sorting (FACS) profiles (counts versus FLH-1 – CyflowCube 6) are shown and the mean fluorescence values from A20 samples ± SEM are represented in the histograms (n=3). Unpaired two-tailed

*Figure 4 continued on next page*

*Figure 4 continued*

Student's t-test was used and asterisks indicate that the mean is significantly different; p-values <0.05 were considered statistically significant. p-Values <0.05 and <0.001 are indicated with * and ***, respectively.

The online version of this article includes the following source data for figure 4:

**Source data 1.** Raw data of flow cytometry analysis of GFP-A20_WT and GFP-tagged A20 variants (Experiment 1).

**Source data 2.** Raw data of flow cytometry analysis of GFP-A20_WT and GFP-tagged A20 variants (Experiment 2).

**Source data 3.** Raw data of flow cytometry analysis of GFP-A20_WT and GFP-tagged A20 variants (Experiment 3).

phenol group of Tyr243 induces substantial change of amino acid interactions (*Figure 5D*). Furthermore, when comparing the alterations of amino acid interactions caused by both p.(Leu236Pro) and p.(Glu192Lys), we observe that more robust interactions exist between Pro236 and the cycle of Phe197 as compared to the Leu236 in the WT counterpart (*Figure 5B and E*); these altered interactions could impact those established between Phe197 and surrounding residues, notably in the region 194–196 that are close to and interact with Glu192 (*Figures 4C, 5E and F*). As for the comparison of the disruptions induced by p.(Leu236Pro) and p.(Cys243Tyr), we first observe that the interactions with residues Asn201 and Tyr306 are lost in A20_Leu236Pro as compared to A20_WT (*Figure 5B and E*) and that interestingly, both Asn201 and Tyr306 do not interact with the WT Cys243 but with variant Tyr243 (*Figure 5D and E*). Similarly, Trp238 that interacts with Leu236 or Pro236 can establish interactions with Tyr243 but not the WT Cys243 (*Figure 5B, D and E*). Lastly, since the amino acid in position 235 is a proline, the p.(Leu236Pro) variation results in two consecutive proline residues, which is uncommonly encountered in proteins compared to singlet prolines (*Morgan and Rubenstein, 2013*) and could hinder proper interactions in this region. For instance, the proper interaction established between Cys243 and Pro235 could be altered by the presence of a proline residue in position 236 as shown in *Figure 5D and E*.

This model suggests the existence, on the properly folded protein, of a critical region of the OTU domain encompassing amino acids affected by pathogenic variations and that are distant on the primary sequence: Glu192, Leu236, and Cys243 (*Figure 5G*).

## Discussion

The diagnosis of HA20 is highly challenging due to its heterogeneous clinical presentation and the lack of pathognomonic symptoms. The non-ambiguous pathogenic significance of *TNFAIP3* variations is necessary for the confirmation of HA20 diagnosis and adaptation of the clinical care. While in the case of *TNFAIP3* nonsense or frameshift variations the deleterious impact is most often established, the pathogenic effect of missense variations is difficult to determine.

Herein, we studied a family in which three patients had a disease phenotype compatible with the diagnosis of HA20 and in whom we identified a novel *TNFAIP3* missense variation, the c.707T>C, p.(Leu236Pro) variation. In the current work, we provide solid proof for the pathogenicity of this variation through an in-depth study relying on the analysis of the clinical and biological features of the patients, the intrafamilial segregation of the variation with the disease, the absence of occurrence of the identified variation in the general population, the conservation of the affected residue, and most importantly, on the functional evaluation of the variant through a combination of different assays: expression and stability of the variant protein, study of its ability to inhibit the NF-κB pathway and of its deubiquitination activity. By assessing, through dedicated in silico and in vitro assays, the pathogenicity of other missense variations located in the OTU domain of A20 and that fulfill the same criteria, we identified a critical region in the OTU domain in which residues involved in pathogenic missense variations establish common interactions.

The clinical features and the cytokine profile of the patients presented in our study are in line with those exhibited by previously reported HA20 patients. Indeed, elevated levels of pro-inflammatory cytokines in the plasma and supernatants of LPS-stimulated PBMCs of the patients presented in our study resemble that of previously reported patients harboring *TNFAIP3* pathogenic truncating variations (*Rajamäki et al., 2018*; *Zhou et al., 2016*). Accordingly, the patients reported herein exhibit typical HA20 phenotypic features: early-onset, recurrent fever, articular symptoms, recurrent oral, genital and/or gastrointestinal manifestations. Nevertheless, HA20 clinical manifestations can vary

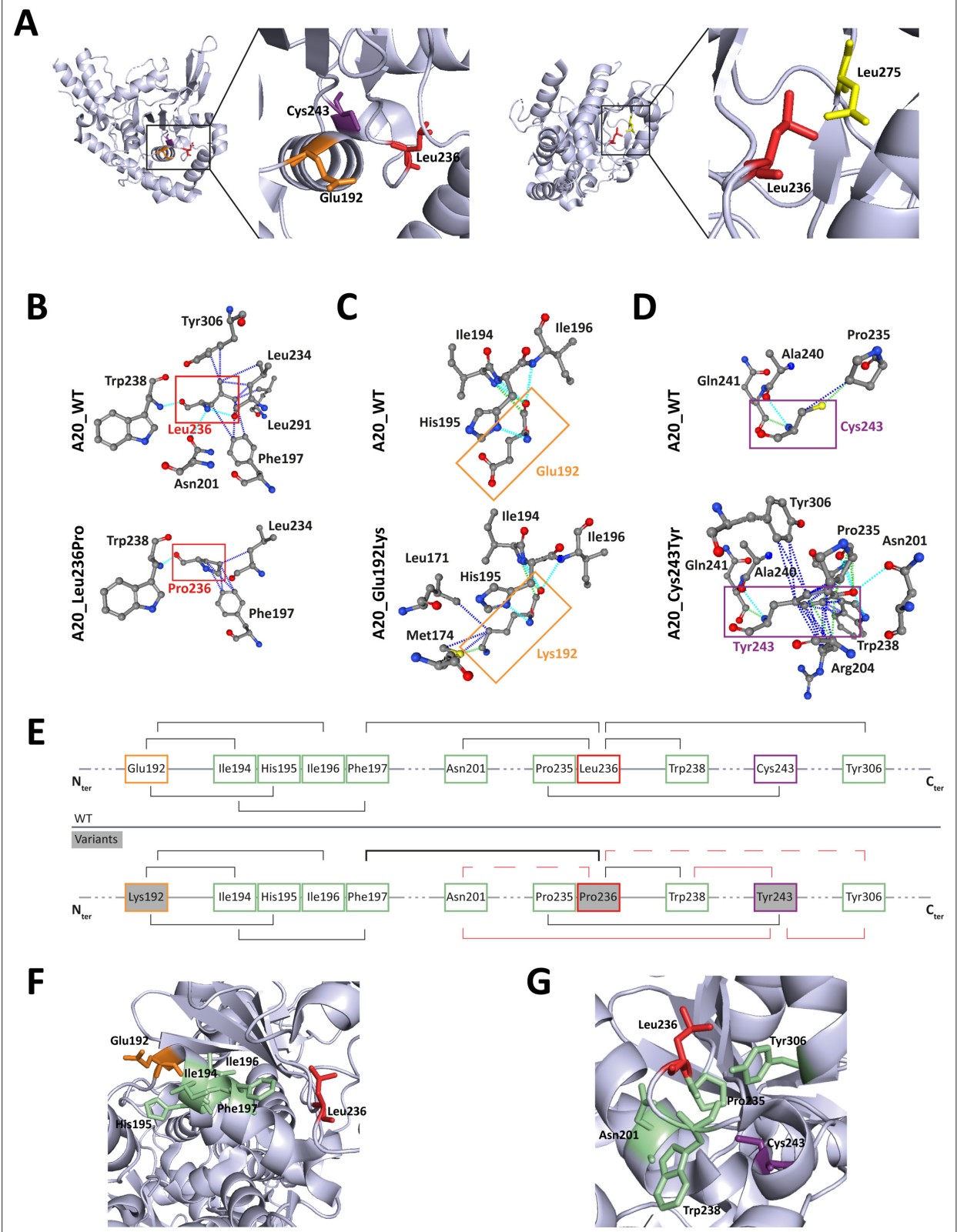

**Figure 5.** Localization and interactions of amino acids involved in A20 variants. (**A**) Amino acids Leu236 (red), Glu192 (orange), and Cys243 (purple) are placed on the crystal structure of the ovarian tumor (OTU) domain of A20 using the 3zjd crystal available of PDB and PyMol software. (**B, C, D**) Prediction of the amino acid interactions in A20_WT or A20 variant proteins established by (**B**) Leu236 (upper panel) or Pro236 (lower panel), (**C**) Glu192 (upper panel) or Lys192 (lower panel), (**D**) Cys243 (upper panel) or Tyr243 (lower panel), using the 3zjd crystal available of PDB and the PremPS in silico

*Figure 5 continued on next page*

*Figure 5 continued*

prediction tool. Oxygen atoms are represented in red; nitrogen atoms are represented in blue; the remaining atoms are in gray. The interactions are shown in dotted lines: hydrophobic (dark blue), polar (turquoise blue), and Van der Waals (green). (**E**) Schematic representation of the predicted amino acid interactions in A20_WT (upper panel) and the variant protein (lower protein). The amino acids involved in pathogenic variations are: Glu192 (orange frame), Leu236 (red frame), and Cys243 (purple frame). On the lower panel, the variant counterpart is represented in gray background and changes in amino acid interactions compared to A20_WT are represented in red: full lines represent newly formed interaction, dotted lines represent the loss of an interaction compared to A20_WT, the thicker line represents a higher number of possible interactions. (**F**, **G**) Amino acids are placed on the crystal structure of the OTU domain of A20 using the 3zjd crystal available of PDB and PyMol software: residues not affected by variations are shown in light green, Glu192 is shown in orange, Leu236 in red. and Cys243 in purple. In (**F**), the helix formed by amino acids 111–149 is hidden to allow for better graphical representation of the amino acids of interest.

considerably among patients even among those who carry the same variation (*Aeschlimann and Laxer, 2019*). Depending on disease severity, colchicine treatment, corticosteroids, or cytokine antagonists can be required for HA20 treatment. Colchicine treatment was satisfactory for the control of the symptoms of individuals II.1 and II.2, suggesting a less severe disease presentation, whereas corticosteroids and azathioprine were required for individual I.1. The difference in disease severity observed in these patients cannot be attributed to the missense nature of the variation. In fact, other HA20 patients carrying pathogenic missense variations were shown to be resistant to colchicine and required steroid therapy (*Dong et al., 2019*; *Kadowaki et al., 2021*; *Shigemura et al., 2016*). Disease management of the patients reported in our study did not require cytokine antagonists. Nevertheless, in light of the elevated IL6 levels in PBMCs supernatant prior to LPS stimulation, the use of IL6 antagonist for the control of the patients' symptoms could be of therapeutic interest in case the patients develop resistance to their current treatments, as previously reported for the treatment of two HA20 patients (*Lawless et al., 2018*; *Zhou et al., 2016*).

Further findings supporting the pathogenic significance of the p.(Leu236Pro) variation include enhanced UPS-dependent degradation of the A20_Leu236Pro protein, which is most likely responsible for the HA20 observed in the patients' PBMCs. We show that this abnormally enhanced degradation is also observed for another missense variant (p.(Leu275Pro), *Aslani et al., 2022*) for which functional assays have not been reported yet. Interestingly, the in silico tools predicted high destabilizing effects for both A20_Leu236Pro and A20_Leu275Pro. Altogether, the flow cytometry-based functional assay, in combination with the in silico approach, identifies enhanced recognition by the protein quality control machinery and thus increased degradation as one of the mechanisms explaining the pathogenicity of TNFAIP3 missense variants. Our study thus provides a valuable approach that can be used for assessing newly identified missense variations.

Moreover, our experiments indicate that, in addition to reduced levels of the variant protein, the function of A20_Leu236Pro is altered. Indeed, in an attempt to bypass the decreased protein expression in vitro, we doubled the amounts of mutant A20 plasmid used for transfection in the NF-κB luciferase assay. Although this allowed for an increased inhibition of the canonical NF-κB pathway, the inhibitory function of A20_Leu236Pro was not restored up to the levels of the WT protein. Interestingly, such functional deregulation has also been observed for the pathogenic p.(Cys243Tyr) and p.(Glu192Lys) variants (*Kadowaki et al., 2021*; *Shigemura et al., 2016*): a 10-fold protein expression increase was required to compensate the loss-of-function of the p.(Cys243Tyr) variant (*Shigemura et al., 2016*). As for the p.(Glu192Lys) variant, it does not exhibit reduced expression compared to the WT protein but it loses the ability to sufficiently inhibit the NF-κB pathway (*Kadowaki et al., 2021*), in line with its location at the putative ubiquitin-binding site of A20 (*Lin et al., 2008*).

The consequences on the native structure of A20 of the three pathogenic missense variations of the OTU domain (p.(Leu236Pro), p.(Glu192Lys), and p.(Cys243Tyr)) converge. Although Glu192 and Cys243 are distant, Leu236 bridges the interactions between the two regions they belong to on the tertiary structure of A20. Moreover, p.(Leu236Pro) and p.(Cys243Tyr) have common consequences on interactions established by surrounding amino acids such as Asn201, Pro235, Trp238, and Tyr306, suggesting a putative importance of these residues for proper protein expression and/or function.

The OTU domain of A20 harbors the active site essential for the DUB activity of A20, which encompasses the catalytic Cys103, His256, and a third residue for which Asp70 or Thr97 are possible candidates (*Komander and Barford, 2008*; *Lin et al., 2008*). Study of the crystal structure of the A20 OTU domain shows that the ubiquitin-binding site of A20 may require the surface formed by α-helices

(α4, α7, and α8) and β sheets (β2, β3, β4, and β5) and the loops between them; Glu192 is among the surface residues identified in the ubiquitin-binding region, whereas Leu236 and Cys243 are in the loop between β3 and β4 (*Lin et al., 2008*). Ubiquitin binding requires interactions between hydrophobic residues of ubiquitin and hydrophobic A20 amino acid, making Leu236 a very suitable candidate for surface amino acids in the ubiquitin-binding domain. Overall, the HA20-causing missense variations in the OTU domain, which alter the proper function and/or expression of the protein, are located in a critical region outside the DUB active site but close to the ubiquitin-binding domain.

## Conclusion

In the present study, we analyzed a family with three patients presenting with a phenotype reminiscent of HA20 in whom we identified the p.(Leu236Pro) missense variation in the *TNFAIP3* gene that segregates with the disease. We provide evidence from patient-derived samples for the HA20, as well as cellular evidence for the increased proteasomal degradation and the loss-of-function effect induced by this variation. By studying the pathogenic missense *TNFAIP3* variations located in the OTU domain of the protein, we shed the light on a non-previously identified pathogenic missense variation-containing region. Our data can help in the interpretation of *TNFAIP3* missense variations and, in turn, in providing an adapted clinical care for the patients.

# Materials and methods
## Affected and healthy individuals

Clinical features were collected through a standardized form. Written informed consent and consent to publish were obtained from each individual for genetic tests according to French legislation and the principles of the Declaration of Helsinki regarding ethical principles for medical research. This study is part of the research projects approved by the ethics evaluation of the Institut national de la santé et de la recherche médicale (No: 21-851).

## Next-generation sequencing

NGS was performed using a custom sequence capture (Nimblegen SeqCap EZ Choice system; Roche Sequencing, Pleasanton, CA, USA) of the exons and the flanking intronic sequences of the main auto-inflammatory disease-causing genes. Sequencing was performed on Nextseq500 platform (Illumina, San Diego, CA, USA) according to the manufacturer's instructions. Variations and segregation analysis were confirmed by standard PCR and Sanger sequencing of DNA extracted from peripheral blood.

## Sanger sequencing

*TNFAIP3* was amplified by PCR and sequenced using the Big-Dye Terminator sequencing kit (Applied Biosystems, Foster City, CA, USA) on an ABI3730XL automated capillary DNA sequencer (Applied Biosystems). Sequences were analyzed against the reference sequence (NM_006290) using the SeqScape software (Applied Biosystems).

## Plasma and PBMC isolation

Whole blood samples from patients and healthy individuals (provided by the Etablissement Français du Sang) were collected in EDTA tubes. Plasma was isolated by blood centrifugation at 950 × *g* for 10 min and kept at −80°C for cytokine measurements. For PBMC isolation, blood samples were diluted with an equal volume of PBS-EDTA (1 mM) and PBMCs were collected by centrifugation (800 × *g*, 15 min) of the diluted samples on Pancoll gradient tubes (PANbiotech, Aidenbach). PBMCs were washed three times with PBS-EDTA and incubated at 37°C for 1 hr in RPMI medium before being treated with LPS (100 ng/mL for 16 hr) before proceeding to ELISA cytokine quantification.

## ELISA cytokine quantification

IL1β, IL18, IL6, TNFα, IFNγ, IP10, and IL10 quantification was carried out on plasma and PBMC culture medium using R&D systems kits (Minneapolis, MN, USA) according to the manufacturer's instructions. For each cytokine quantification, samples from at least five healthy individuals were used as controls. At the time of sample collection, individual I.1 suffered from arthralgia and presented ulcers while individuals II.1 and II.2 were in remission.

## Plasmids

Full-length human *TNFAIP3, TRAF6,* and *Ubiquitin-K63* were purchased from Addgene pEGFP-C1-A20 – Plasmid #22141, pcDNA3 Flag-humanTRAF6 (1093) Addgene #66929, pRK5-HA-Ubiquitin-K63 Addgene #17606. The empty vector pEGFP-C1 was a gift from Dr. Frédéric Lezot. Directed mutagenesis using the Q5 High-Fidelity DNA Polymerase (New England Biolabs, Ipswich, MA, USA) was used.

## Cell lines

HEK293T (human embryonic kidney 293T) were originally purchased from the American Type Culture Collection (ATCC). Mycoplasma contamination testing was performed regularly using the VenorGeM OneStep mycoplasma contaminations detection kit.

## Cell culture and transfection

HEK293T cells were maintained in DMEM containing GlutaMAX (Thermo Fisher Scientific, Waltham, MA, USA) supplemented with 10% fetal bovine serum and 1% penicillin/streptomycin (Thermo Fisher Scientific) at 37°C in a 5% $CO_2$ atmosphere. Cells were transiently transfected using Fugene Transfection Reagent (Promega, Madison, WI, USA) according to the manufacturer's instructions.

## Proteasome inhibition

HEK293T cells were transfected with the indicated plasmids (GFP-tagged *TNFAIP3* constructs) and treated 24 hr later with 20 µM MG132 (Sigma-Aldrich, St. Louis, MO, USA) or the equivalent volume of DMSO (Sigma-Aldrich) for 5 hr. Proteins were then extracted and detergent-soluble and insoluble fractions of protein lysates were analyzed by western blot. Alternatively, cells were collected, washed, and centrifuged at 200 × *g* for 5 min. Cell pellets were resuspended in BSA 1% solution and sorted by FACS (fluorescence-activated cell sorting). The intensity of GFP signal per cell was determined by flow cytometry using the CyView software.

## Cycloheximide study

HEK293T cells were transfected with the indicated plasmids, and treated 24 hr later with 100 µg/mL cycloheximide (Sigma-Aldrich) or the equivalent volume of DMSO for the indicated times.

## Western blot analysis

Protein extraction and western blot analysis were carried out as previously described (*El Khouri et al., 2021*). The primary antibodies used were: polyclonal rabbit A20 (Sigma-Aldrich HPA018846), anti-GFP-HRP (Cell Signaling 2037), anti-HA (Sigma-Aldrich H3663), anti-FLAG (Sigma-Aldrich F1804), anti-GAPDH-HRP (Cell Signaling 51332), and anti-α-Tubulin-HRP (Cell Signaling 9099). The secondary antibodies used were Anti-Mouse IgG-HRP, Sigma-Aldrich A054 and Anti-Rabbit IgG-HRP, Sigma-Aldrich A0545. Proteins were detected with Amersham ECL Select Western Blotting Detection Reagent (GE Healthcare, Chicago, IL, USA) and Bio-Rad ChemiDoc Imaging Systems was used for detection. The ImageJ software was used for signal quantification.

## NF-κB luciferase assay

HEK293T cells were transiently transfected with the indicated quantities of *TNFAIP3* WT and mutant plasmids, 100 ng of pGL4.32[luc2P/NF-κB-RE/Hygro] vector and 10 ng of Renilla Luciferase Control Reporter Vector (pGL4-hRluc), purchased from Promega. 24 hr later, cells were treated with 10 ng/mL TNFα for 5 hr then lysed with Passive Lysis Buffer (Promega). Firefly luciferase activity was measured in duplicate according to the manufacturer's instructions and normalized against the Renilla signal.

## Deubiquitination assay

HEK293T cells were transfected with the indicated plasmids. Forty-eight hours post transfection, cells were collected and proteins were extracted as previously described (*El Khouri et al., 2021*). Immunoprecipitation was conducted on protein lysates using the ANTI-FLAG M2 Magnetic Beads (#M8823, Sigma-Aldrich), according to the manufacturer's instructions. Briefly, equal amounts of detergent-soluble proteins were incubated with the resin beads overnight with shaking at 4°C. The beads were then washed three times with Tris Buffered Saline solution. Elution was performed using sample buffer by incubation for 10 min at 95°C. Eluted proteins were then analyzed by western blot.

## Crystal structure and amino acid interaction analysis

The PremPS (Predicting the Effects of Mutations on Protein Stability) tool, a computational method that predicts protein stability changes induced by missense variations available as a web server (https://lilab.jysw.suda.edu.cn/research/PremPS/) (*Chen et al., 2020b*), was used to predict protein stability and to analyze amino acid interactions of the Protein Data Bank-available crystal structure of A20 OTU domain in reduced, active state at 1.87 Å resolution (PDB accession number 3ZJD) (*Kulathu et al., 2013*). The amino acid interactions were shown as provided by the PremPS online tool. Three-dimensional structure was visualized and amino acids were positioned using the PyMol Software.

## Identification of *TNFAIP3* missense variations reported in the literature

Literature review of *TNFAIP3* variations was performed in PubMed using the terms "HA20", 'A20', or 'TNFAIP3' and "variation" or 'mutation'.

## Statistical analyses

Statistical analyses and graph representation were performed using GraphPad Prism 9.0 software. Unpaired two-tailed Student's t-tests were used for comparisons of the means of two groups. The means of the data provided by sets of independent experiments carried out on samples from the patients (or cells expressing the variant protein) were compared to the means of data provided by sets of independent experiments carried out on the samples from healthy individuals (or cells expressing the WT protein). Data are presented as means ± SEM and individual data points are represented when required. p-Values <0.05 were considered statistically significant. p-Values <0.05, <0.01, <0.001, and <0.0001 are indicated with *, **, ***, and ****, respectively.

## Acknowledgements

We thank the patients and their family as well as the control individuals for their cooperation. The research reported in this manuscript was funded by: Institut National de la Santé et de la Recherche Médicale (INSERM); Agence Nationale de la Recherche (ANR-17-CE17-0021-01); H2020, European Union, Grant Agreement no. 779295; and Sorbonne Université EMERGENCE-PhenomAID.

## Additional information

### Funding

| Funder | Grant reference number | Author |
|---|---|---|
| Institut National de la Santé et de la Recherche Médicale | 00552404 | Camille Louvrier |
| Agence Nationale de la Recherche | ANR-17-CE17-0021-01 | Serge Amselem |
| H2020 European Research Council | 779295 | Serge Amselem |
| Sorbonne Université | Émergence PhenomAID | Irina Giurgea |

The funders had no role in study design, data collection and interpretation, or the decision to submit the work for publication.

### Author contributions

Elma El Khouri, Conceptualization, Data curation, Formal analysis, Validation, Investigation, Visualization, Methodology, Writing – original draft; Farah Diab, Conceptualization, Formal analysis, Investigation, Writing – original draft; Camille Louvrier, Eman Assrawi, Conceptualization, Formal analysis, Methodology, Writing – review and editing; Aphrodite Daskalopoulou, Investigation; Alexandre Nguyen, Alexandra Desdoits, Resources; William Piterboth, Formal analysis, Investigation; Samuel Deshayes, Resources, Writing – review and editing; Bruno Copin, Software, Methodology; Florence Dastot Le Moal, Investigation, Methodology; Sonia Athina Karabina, Conceptualization,

Funding acquisition, Methodology, Writing – review and editing; Serge Amselem, Conceptualization, Resources, Data curation, Supervision, Funding acquisition, Methodology, Project administration, Writing – review and editing; Achille Aouba, Conceptualization, Resources, Writing – review and editing; Irina Giurgea, Conceptualization, Resources, Data curation, Supervision, Funding acquisition, Validation, Methodology, Project administration, Writing – review and editing

## Author ORCIDs

Irina Giurgea (ID) http://orcid.org/0000-0002-5035-2958

## Ethics

Clinical features were collected through a standardized form. Written informed consent and consent to publish were obtained from each individual for genetic tests according to French legislation and the principles of the Declaration of Helsinki regarding ethical principles for medical research. This study is part of the research projects approved by the ethics evaluation of the Institut national de la santé et de la recherche médicale (No: 21-851).

## Decision letter and Author response

Decision letter https://doi.org/10.7554/eLife.81280.sa1
Author response https://doi.org/10.7554/eLife.81280.sa2

---

# Additional files

## Supplementary files

• MDAR checklist

## Data availability

Figure 3 - Source Data 1 to 5 contain the original western blots used for figure 3Figure 4 - Source Data 1 to 3 contain the original flow cytometry original data used for figure 4.

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
