## [Editor Report]

The study is valuable to clinical diagnostic laboratories and clinicians as it investigates the functional significance of missense variants in the TNFAIP3 gene encoding A20. Haploinsufficiency of A20 (HA20) is a dominantly inherited disease resulting from monoallelic loss-of-function mutations, while the contribution of novel/rare missense variations is unclear, and they are classified as variants of unknown significance (VUS). The authors developed a new flow cytometry functional assay that other laboratories can adapt to evaluate the pathogenicity of A20 VUSs.

---

## [Decision Letter]

**Decision letter after peer review:**

Thank you for submitting your article "Identification of an A20 critical region harboring missense variations that lead to autoinflammation" for consideration by *eLife*. Your article has been reviewed by 3 peer reviewers, and the evaluation has been overseen by a Reviewing Editor and Carla Rothlin as the Senior Editor. The following individual involved in review of your submission has agreed to reveal their identity: Daniella Schwartz (Reviewer #1).

Essential revisions:

The three reviewers and the guest reviewing editor agreed that the authors need to do additional experiments to support their conclusions on the effect of the novel p.L236P and other HA20-associated missense variants on protein function:

1. The authors should demonstrate using in vitro Ub-binding experiments in transfected cells that the mutant L236P protein fails to bind Ub proteins. Alternatively, the DUB activity of the mutant protein could be tested both in vitro and in the patient's primary cells.

2. The assays showing the reduced stability of p.L236P A20 compared to WT A20 are convincing. The authors should perform this assay on at least 1-2 other pathogenic missense mutations and one common benign missense variant in the OTU domain.

3. Additional in silico evaluation is recommended for a few other missense mutations, including recently reported L83F, V377I, W356R, and T602S. These variants should be added to Table 1.

*Reviewer #1 (Recommendations for the authors):*

1) Effects of mutation on protein folding.

The authors present some very nice in silico data; it would be helpful if this data were supported by biochemical folding assays using protein carrying this mutation (p.Leu236Pro) and others in the OTU domain (pathogenic and nonpathogenic) to show that the pathogenic mutations specifically affect protein folding

2) Generalizability of destabilizing mutations to other OTU mutations

The assays showing the reduced stability of p.Leu236Pro A20 and WT A20 (half life, cyclohexamide and MG132 assay) are convincing and well done. To further support the generalizability of this mechanism, it would be helpful to perform this assay on at least 1-2 other pathogenic missense mutations in the OTU domain as well as a benign variant.

3) Missed pathogenic variants.

On my review of the literature (pubmed and infevers.com) I identified several pathogenic or likely pathogenic mutations that were not evaluated by the group. These included:

The following missense mutations in the OTU domain

– L83F

– W356R

And missense outsider the OTU domain

– T602S

These mutations should be in Table 2 and should also be included in the in silico analysis.

*Reviewer #2 (Recommendations for the authors):*

1. This is a consanguineous family, please show the homozygous gene list, to confirm that no other additional homozygous mutation contributes to the phenotype.

2. Figure 1A, show the sanger sequencing results of patients I.1 and II.1.

3. Figure 2, are the cytokine levels of the patients during flares or during remission?

4. The mutant Leu236Pro A20 has decreased stability, is it possible to explain the mechanisms why the mutation resulted in enhanced UPS dependent degradation?

5. In addition to luciferase assay, please also perform more functional studies to demonstrate the loss-of-function of the variant, such as reduced de-ubiquitination in the mutant protein.

*Reviewer #3 (Recommendations for the authors):*

I think it may be suitable for a specific clinical journal due to the lack of new findings.

---

## [Author Response]

Essential revisions:The three reviewers and the guest reviewing editor agreed that the authors need to do additional experiments to support their conclusions on the effect of the novel p.L236P and other HA20-associated missense variants on protein function:1. The authors should demonstrate using in vitro Ub-binding experiments in transfected cells that the mutant L236P protein fails to bind Ub proteins. Alternatively, the DUB activity of the mutant protein could be tested both in vitro and in the patient's primary cells.

As requested, we have conducted additional experiments to assess the deubiquitinase (DUB) function of our mutant. As we no longer have protein samples derived from patients’ PBMCs, we have co-expressed A20 WT or A20 L236P constructs together with the A20 target TRAF6 and K63-linked ubiquitin in HEK293T cells and evaluated the ubiquitination state of TRAF6. This co-immunoprecipitation experiment revealed an increased intensity of the high molecular weight ubiquitin-aggregates smear of TRAF6 in the presence of A20_Leu236Pro compared to the WT condition, thus indicating an altered DUB activity of the A20_Leu236Pro. This experiment has now been described in the Results section (lines 168 to 174 in the Results section) and is represented in a new figure (Figure 3F).

2. The assays showing the reduced stability of p.L236P A20 compared to WT A20 are convincing. The authors should perform this assay on at least 1-2 other pathogenic missense mutations and one common benign missense variant in the OTU domain.

In order to test whether other A20 missense variations than the p.(Leu236Pro) variant also exhibit enhanced proteasomal degradation, we have quantified the levels of A20 proteins upon MG132-induced proteasome inhibition in HEK293T cells. In order to test whether TNFAIP3 missense variations other than the p.(Leu236Pro) variant exhibit enhanced proteasomal degradation, we sought to quantify the levels of several A20 proteins, each protein carrying different reported missense variations upon MG132-induced proteasome inhibition in HEK293T cells. For this purpose, we took advantage of the GFP tag present in the TNFAIP3 constructs to sort the cells by flow cytometry and quantify the intensity of the GFP fluorescent signal in cells treated with MG132 or DMSO as negative control. For this analysis, in addition to A20_WT and A20_Leu236Pro, we included the likely pathogenic Leu275Pro variant, described in Aslani et al. (Aslani et al., 2022) because the in silico PremPS tool predicted a high destabilizing effect of the variation (ΔΔG = 1.80 kcal mol-1 – Table 2) and Leu275Pro protein expression has not been assessed to date. As a positive control for the experiment, we used the construct coding for A20_Leu277* truncated protein (Zhou et al., 2016) which should be recognized by the protein quality control as a misfolded protein. We also chose to include the p.(Asn102Ser) variant, previously suggested to lead to HA20 (Yu Chen et al., 2020). This variant is, however, frequent in the control populations (allele frequency of 0.01236 if considering all populations and of 0.06389 in the Latino population according with gnomAD) and has a low destabilizing predicted effect (ΔΔG = 0.78 kcal mol-1 – Table 2). As a negative control, we chose the polymorphic variant p.(Phe127Cys) (allele frequency = 0.0617 in gnomAD). Fluorescence intensity increased for A20_Leu236Pro, A20_Leu275Pro and the positive control A20_Leu277*, indicating that like A20_Leu236Pro, the A20_Leu275Pro variant exhibits enhanced proteasomal degradation (curve shifted to the right in Figure 4 upper panel and increased fluorescence intensity in Figure 4 lower panel). This observation is specific to pathogenic variants as no fluorescence intensity shift was observed for the non-pathogenic A20_Phe127Cys variant nor for the A20_Asn102Ser variant. Indeed, these results demonstrated the absence of enhanced proteosomal degradation of the A20_Asn102Ser protein allowing to classify this variant as non-pathogenic (Figure 4).

These results have now been added to Results section (lines 217 to 240). They are described in a new figure (Figure 4) and mentioned in the Discussion section (lines 321 to 328).

3. Additional in silico evaluation is recommended for a few other missense mutations, including recently reported L83F, V377I, W356R, and T602S. These variants should be added to Table 1.

We have analyzed additional missense variations and added them to table 2 (table 1 corresponds to the phenotypic features of our patients). These include the T602S variation mentioned by the reviewer as well as the V377M variation (we indeed found the report of a variation involving V377, but it was V377M and not V377I). We have also added all other missense variations that were recently reported in HA20 patients (i.e. T129M, L275Pro and A547T). As for the L83F and the W356R variations, despite their listing in the database Infevers, we did not find the corresponding original data; they were therefore not included in the analysis. We have included the PremPS *in silico* stability prediction whenever it was possible (presence of the residue in the OTU domain and presence of the residue on the available PDB crystal #3ZJD that we used for the analysis). The manuscript has been modified accordingly (mentioned in blue throughout lines 176 to 215).

Reviewer #1 (Recommendations for the authors):1) Effects of mutation on protein folding.The authors present some very nice in silico data; it would be helpful if this data were supported by biochemical folding assays using protein carrying this mutation (p.Leu236Pro) and others in the OTU domain (pathogenic and nonpathogenic) to show that the pathogenic mutations specifically affect protein folding

We do agree that biochemical folding assays would provide an additional element to the set of evidence we provide to the destabilizing effect of the L236P variation. In our study, the folding defect is not directly assessed but strongly supported by the body of evidence we provide: we first observed reduced steady-state protein levels upon transient expression in HEK293T cells which led us to refer to *in silico* stability prediction tools which revealed a highly destabilizing effect of the Leu236Pro variant. Together, these results called for further investigation of the stability and the degradation of the mutant protein. Therefore, we assessed the stability of the protein by determining its half-life upon protein synthesis inhibition with cycloheximide and we evaluated its degradation by the ubiquitin-proteasome system upon proteasome inhibition with MG132. Overall, these experiments converge towards the targeting of the mutant protein to degradation by the protein quality control system due to its inability to achieve the native state. We hope reviewer #1 finds them convincing as we were unfortunately not able to conduct direct proteins folding assays as they require biophysical expertise or radiolabeling experiments that we do not have access to in our facilities.

As requested for other OTU domain mutants, we have now included the stability prediction for all OTU-domain variants (Table 2) as well as proteasome-degradation assessing for 3 other OTU-domain variants. These results have now been added to the Results section (lines 217 to 240). They are described in a new figure (Figure 4) and mentioned in the Discussion section (lines 321 to 328).

2) Generalizability of destabilizing mutations to other OTU mutationsThe assays showing the reduced stability of p.Leu236Pro A20 and WT A20 (half life, cyclohexamide and MG132 assay) are convincing and well done. To further support the generalizability of this mechanism, it would be helpful to perform this assay on at least 1-2 other pathogenic missense mutations in the OTU domain as well as a benign variant.

In order to test whether other A20 missense variations than the A20_Leu236pro variant also exhibit enhanced proteasomal degradation, we have quantified the levels of A20 proteins upon MG132-induced proteasome inhibition in HEK293T cells. In order to test whether TNFAIP3 missense variations other than the Leu236Pro variant exhibit enhanced proteasomal degradation, we sought to quantify the levels of several A20 proteins each proteins carrying different reported missense variations upon MG132-induced proteasome inhibition in HEK293T cells. For this purpose, we took advantage of the GFP tag present in the TNFAIP3 constructs to sort the cells by flow cytometry and quantify the intensity of the GFP fluorescent signal in cells treated with MG132 or DMSO as negative control. For this analysis, in addition to A20 WT and A20 Leu236Pro, we included the likely pathogenic Leu275Pro variant, described in Aslani et al. because the in silico PremPS tool predicted a high destabilizing effect of the variation (ΔΔG = 1.80 kcal mol-1 – Table 2) and Leu275Pro protein expression has not been assessed to date. As a positive control for the experiment, we used the construct coding for A20_Leu277* truncated protein (Zhou et al., 2016) which should be recognized by the protein quality control as a misfolded protein. We also chose to include the p.(Asn102Ser) variant, previously suggested to lead to HA20 (Yu Chen et al., 2020), but is frequent in the control populations (allele frequency of 0.01236 if considering all populations and of 0.06389 in the Latino population according with gnomAD) and with low destabilizing predicted effect (ΔΔG = 0.78 kcal mol-1 – Table 2). As a negative control, we chose the polymorphic variant p.(Phe127Cys) (allele frequency = 0.0617 in gnomAD). Fluorescence intensity increased for A20_Leu236Pro, A20_Leu275Pro and the positive control A20_Leu277*, indicating that like Leu236Pro, the Leu275Pro variant exhibits enhanced proteasomal degradation (curve shifted to the right in Figure 4 upper panel and increased fluorescence intensity in Figure 4 lower panel). No fluorescence intensity shift was observed for the non-pathogenic Phe127Cys variant nor for the Asn102Ser variant. Indeed, these results demonstrated the absence of enhanced proteosomal degradation of the A20_Asn102Ser protein allowing to classify this variant as non-pathogenic (Figure 4).

These results have now been added to Results section (lines 217 to 240). They are described in a new figure (Figure 4) and mentioned in the Discussion section (lines 321 to 328).

3) Missed pathogenic variants.On my review of the literature (pubmed and infevers.com) I identified several pathogenic or likely pathogenic mutations that were not evaluated by the group. These included:The following missense mutations in the OTU domain– L83F– W356RAnd missense outsider the OTU domain– T602SThese mutations should be in Table 2 and should also be included in the in silico analysis.

We have analyzed additional missense variations and added them to table 2 (table 1 corresponds to the phenotypic features of our patients). These include the T602S variation mentioned by the reviewer. We have also added all other missense variations that were recently reported in HA20 patients (i.e. T129M, L275Pro, V377M and A547T). As for the L83F and the W356R variations, despite their listing in the database Infevers, we did not find the corresponding original data; they were therefore not included in the analysis. We have included the PremPS *in silico* stability prediction whenever it was possible (presence of the residue in the OTU domain and presence of the residue on the available PDB crystal #3ZJD that we used for the analysis). The manuscript has been modified accordingly (mentioned in blue throughout lines 176 to 215).

Reviewer #2 (Recommendations for the authors):1. This is a consanguineous family, please show the homozygous gene list, to confirm that no other additional homozygous mutation contributes to the phenotype.

We have now included the list of genes that we screen in our custom sequence capture approach and that could be responsible for autosomal recessive auto-inflammatory diseases (*ADA2, IL10, IL10RA, IL10RB, IL1RN, IL36RN, IL6ST, LACC1, LPIN2, MEFV, MVK, NLRP1, OTULIN, TRNT1*). We found no mutations in these genes. This information has been added to the Results section (lines 100 to 102).

2. Figure 1A, show the sanger sequencing results of patients I.1 and II.1.

Sanger sequencing of all family members has now been added in supplementary figure S1.

3. Figure 2, are the cytokine levels of the patients during flares or during remission?

At the time of sample collection, Individual I.1 suffered from arthralgia and presented ulcers while individuals II.1 and II.2 were in remission. This information has now been added to the Figure Legend.

4. The mutant Leu236Pro A20 has decreased stability, is it possible to explain the mechanisms why the mutation resulted in enhanced UPS dependent degradation?

Although the identity of the elements specifically involved in A20 protein quality control is, to the best of our knowledge, unknown, the overall mechanisms leading to enhanced UPS-dependent degradation of proteins exhibiting decreased stability rely on the well-established protein quality control system. Indeed, proteins that are unable to achieve the native state – due to a missense variation for instance – are recognized as misfolded and subsequently targeted to degradation. This protein quality control system is composed of molecular chaperones and the ubiquitin proteasome system (UPS). For proteins transiting through the endoplasmic reticulum (like A20), several chaperones recognize misfolded proteins and help their retention in the endoplasmic reticulum, allowing only correctly folded proteins to reach the cytosol. Misfolded proteins are instead ubiquitinated by E3 ligases and targeted to degradation by the proteasome.

5. In addition to luciferase assay, please also perform more functional studies to demonstrate the loss-of-function of the variant, such as reduced de-ubiquitination in the mutant protein.

As requested, we have conducted additional experiments to assess the deubiquitinase (DUB) function of our mutant. As we no longer have protein samples derived from patients’ PBMCs, we have co-expressed A20 WT or A20 L236P constructs together with the A20 target TRAF6 and K63-linked ubiquitin in HEK293T cells and evaluated the ubiquitination state of TRAF6. This co-immunoprecipitation experiment revealed an increased intensity of the high molecular weight ubiquitin-aggregates smear of TRAF6 in the presence of A20_Leu236Pro compared to the WT condition, thus indicating an altered DUB activity of the A20_Leu236Pro. This experiment has now been described in the Results section (lines 168 to 174 in the Results section) and is represented in a new figure (Figure 3F).

Reviewer #3 (Recommendations for the authors):I think it may be suitable for a specific clinical journal due to the lack of new findings.

For the current version of the paper, we have conducted additional experiments to further prove the pathogenicity of the newly identified variant. Furthermore, we have assessed the proteasomal degradation of a series of variants in addition to the one we report in our study. For this purpose, we have generated several additional constructs and developed a new flow cytometry assay (new Figure 4). This functional assay, in combination with a dedicated in silico approach, identifies the enhanced recognition of the mutated proteins by the protein quality control machinery and thus the increased degradation as one of the mechanisms explaining the pathogenicity of TNFAIP3 missense variants. With all the additional experiments we have performed in response to the editor and the reviewers, we hope that Reviewer # 3 is now convinced that the current version of the manuscript indeed provides new findings to the scientific community.